# Relations between Sensory Responsiveness and Features of Autism in Children

**DOI:** 10.3390/brainsci10110775

**Published:** 2020-10-24

**Authors:** Jacob I. Feldman, Margaret Cassidy, Yupeng Liu, Anne V. Kirby, Mark T. Wallace, Tiffany G. Woynaroski

**Affiliations:** 1Department of Hearing & Speech Sciences, Vanderbilt University Medical Center, Nashville, TN 37232, USA; mark.t.wallace@vanderbilt.edu (M.T.W.); tiffany.g.woynaroski@vumc.org (T.G.W.); 2Neuroscience Undergraduate Program, Vanderbilt University, Nashville, TN 37232, USA; margaret.m.cassidy@vanderbilt.edu (M.C.); yupeng.liu@vanderbilt.edu (Y.L.); 3National Institutes of Health, Bethesda, MD 20814, USA; 4Washington University School of Medicine, Washington University in St. Louis, St. Louis, MO 63110, USA; 5Department of Occupational and Recreational Therapies, University of Utah, Salt Lake City, UT 84112, USA; anne_kirby@hsc.utah.edu; 6Vanderbilt Kennedy Center, Vanderbilt University Medical Center, Nashville, TN 37232, USA; 7Vanderbilt Brain Institute, Vanderbilt University, Nashville, TN 37232, USA; 8Frist Center for Autism & Innovation, Vanderbilt University, Nashville, TN 37232, USA; 9Department of Psychology, Vanderbilt University, Nashville, TN 37232, USA; 10Department of Psychiatry and Behavioral Sciences, Vanderbilt University Medical Center, Nashville, TN 37232, USA; 11Department of Pharmacology, Vanderbilt University, Nashville, TN 37232, USA

**Keywords:** autism, sensory responsiveness, language, sensory seeking, hyporesponsiveness, hyperresponsiveness

## Abstract

Autism is a neurodevelopmental condition defined by differences in social communication and by the presence of restricted and repetitive patterns of behavior, interests, and activities (RRBs). Individuals with autism also commonly present with atypical patterns of sensory responsiveness (i.e., hyporesponsiveness, hyperresponsiveness, and sensory seeking), which are theorized to produce cascading effects across other domains of development. The purpose of this study was to examine differences in sensory responsiveness in children with and without autism (ages 8–18 years), as well as relations between patterns of sensory responsiveness and core and related features of autism. Participants were 50 children with autism and 50 non-autistic peers matched on age and sex. A comprehensive clinical battery included multiple measures of sensory responsiveness, core features of autism, adaptive behavior, internalizing behaviors, cognitive ability, and language ability. Groups significantly differed on all three patterns of sensory responsiveness. Some indices of core and related autism features were robustly associated with all three patterns of sensory responsiveness (e.g., RRBs), while others were more strongly associated with discrete patterns of sensory responsiveness (i.e., internalizing problem behaviors and hyperresponsiveness, language and sensory seeking). This study extends prior work to show that differences in sensory responsiveness that are linked with core and related features of autism persist in older children and adolescents on the spectrum.

## 1. Introduction

Autism is a neurodevelopmental condition that has conventionally been defined by differences in social communication and by the presence of restrictive and repetitive patterns of behaviors, interests, and activities (RRBs) [1]. The most recent diagnostic criteria, however, additionally recognize differences in sensory functioning as a diagnostically relevant feature of autism [2], and a growing body of literature shows that individuals with autism (see Appendix A for information on our flexible use of terminology) present with altered patterns of sensory responsiveness compared to non-autistic peers [3,4]. Three patterns of sensory responsiveness are commonly described in the literature: hyperresponsiveness, or exaggerated responding to sensory stimuli; hyporesponsiveness, or reduced responding to sensory stimuli; and sensory seeking, or craving of certain sensory stimuli [4,5,6,7]. Although these classifications may be distinct, they are not mutually exclusive or restricted to one sensory modality; individuals can and do present with varied combinations of alterations across patterns of sensory responsiveness and sensory modalities, e.g., [3,8].

It has been proposed that differences in sensory responsiveness, particularly early in life, may produce cascading effects on development across domains, potentially causing or contributing to characteristics associated with autism [9,10]. Findings from empirical research lend some support to this theory. Atypical patterns of sensory responsiveness, for example, have been linked to the core features of RRBs [6,11,12,13,14,15] and social communication in persons on the autism spectrum [12,16,17,18,19,20,21]. 

In addition to the aforementioned ‘core’ characteristics, individuals with autism may present with a variety of related (i.e., often co-occurring) features that impact their quality of life and long-term outcomes. For example, language abilities in children with autism are highly variable, and early language proficiency is associated with social, academic, and vocational success later in life in this population, e.g., [22,23]. There is evidence to suggest that sensory differences are associated with language abilities, at least in preschoolers with autism [21,24,25]. Other related differences commonly observed in autism, including decreased adaptive behaviors and increased prevalence of anxiety and related behavioral concerns, have also been associated with patterns of sensory responsiveness, e.g., [8,20,25,26,27,28,29,30,31]. 

The aforementioned findings for associations between patterns of sensory responsiveness and core and related characteristics of autism provide some empirical support for the theory of ‘cascading effects’. There are, however, several limitations of the extant literature. First, the majority of studies to date were conducted with young children, though the literature suggests that differences in sensory responsiveness may persist into adolescence and even adulthood in individuals with autism e.g., [32,33,34,35]. Furthermore, the vast majority of past reports have focused on only one aspect of sensory functioning, e.g., [12,20,26], or have explored relations with only one domain of core or related features of autism, e.g., [6,14,20]. Finally, no study to our knowledge has investigated whether any of the three patterns of sensory responsiveness is a value-added predictor of core and related autism characteristics (i.e., whether one pattern of sensory responsiveness accounts for unique variance in autism features of interest after controlling for the other two patterns of sensory responsiveness; [36]).

The current study thus seeks to comprehensively examine sensory responsiveness in an older sample of school-aged children and adolescents with and without autism and to test relations between all three sensory constructs of interest (i.e., hyporesponsiveness, hyperresponisveness, and sensory seeking) with a broad range of core and related features of autism. Specifically, the research questions to be examined are:

Are between-group differences in sensory responsiveness present in children and adolescents with autism and non-autistic peers matched on chronological age and biological sex? 

Are there significant associations between patterns of sensory responsiveness and core and related features of autism? 

What proportion of the variance in core and related features of autism is uniquely accounted for by each of the three patterns of sensory responsiveness? 

Do any of the three patterns of sensory responsiveness account for a significant amount of unique variance in core and related characteristics of autism when controlling for the other two patterns?

## 2. Materials and Methods

All experimental procedures were carried out in compliance with the Vanderbilt University Medical Center IRB. The project was initially approved on 15 September 2010 (project no. 101194).

### 2.1. Participants

Participants were 50 children and adolescents with autism (*M*_age_ = 13.1 years; 37 male; 13 female) and 50 children and adolescents without autism (*M*_age_ = 13.0 years; 37 male; 13 female) matched at the group level on both chronological age and biological sex. Participants were drawn from a larger study of sensory functioning in children with and without autism e.g., [37,38,39]. Nonverbal cognitive ability (NVIQ) was measured by either the Leiter International Performance Scale–Third edition (Leiter–3) or the Test of Nonverbal Intelligence–Fourth edition (TONI–4) [40,41]. NVIQ means for the autism and non-autism groups were 108.32 and 118.13, respectively (see Table 1).

Inclusion criteria for this study were: (a) chronological age between 8;0 and 18;11 years; (b) normal or corrected-to-normal vision and normal hearing, as confirmed by screening at entry to the study; (c) no history of seizure disorders; and (d) no parent report of diagnosed genetic disorders, such as Fragile X or tuberous sclerosis. Additional inclusion criterion for children with autism was a diagnosis of autism according to DSM-5 criteria [2], as confirmed by a research-reliable administration of the Autism Diagnostic Observation Schedule-Second edition (ADOS-2) [42] and the judgement of a licensed clinician on the research team. Additional inclusion criteria for children without autism were (a) scores below the screening threshold for autism risk on the Social Communication Scale (SCQ) [43]; (b) no immediate family members with a diagnoses of autism; (c) NVIQ above 85; and (d) no prior history or present indicators of psychiatric conditions or learning disorders.

### 2.2. Materials

See Table 2 for a summary of measures and variables derived from those measures for use in planned analyses.

#### 2.2.1. Measures of Sensory Responsiveness

Parents of participants completed the Sensory Experience Questionnaire-Version 2.1 (SEQ) and the Sensory Profile (SP) [5,44]. The summary scores indexing hyporesponsiveness, hyperresponsiveness, and sensory seeking from the SEQ and SP (i.e., SP poor registration and SEQ hyporesponsiveness; SP sensory sensitivity, SP sensation avoiding, SEQ hyperresponsiveness; SP and SEQ sensory seeking) were converted to z-scores. All variables purported to measure the same construct were sufficiently intercorrelated to warrant aggregation (|*r*| = 0.48–0.71; note that scores from the SP were reflected such that higher scores indicated higher presence of these features, in accord with SEQ scoring). The z-scores purported to tap the same construct were then averaged to create aggregate scores for hyporesponsiveness, hyperresponsiveness, and sensory seeking [32]. 

#### 2.2.2. Measures of Core Features of Autism

The SCQ, a screening measure used to characterize parents’ concerns related to autism symptomatology, was used to calculate a summary score for the presence of core features of autism. Parents also completed the Social Responsiveness Scale (SRS), a survey instrument that yields indices of RRBs, social communication differences, and total autistic features [45]. As the SCQ scores were highly intercorrelated with the SRS total T scores (*r* = 0.89), an aggregate of autistic features was generated by averaging *z*-scores for the aforementioned component variables. 

#### 2.2.3. Measures of Related Features of Autism

Parents completed the Vineland Adaptive Behavior Scales-Second edition (VABS), a norm referenced clinical instrument used to measure adaptive behavior across the lifespan in domains including communication, daily living skills, and socialization [46]. Standard scores for each domain were used in analyses. 

Participants also completed the Behavior Assessment System for Children-Third edition (BASC), a norm referenced parent/caregiver report measure that assesses child behavioral and emotional functionality [47]. From this assessment, T-scores were extracted in the following domains: anxiety, depression, and somatization, which comprise the internalizing problem behaviors composite, and the functional impairment probability index. 

#### 2.2.4. Measures of Language Ability 

Participants completed a battery of standardized, norm-referenced language assessments, including the Receptive One Word Picture Vocabulary Test-Fourth edition (ROWPVT) and the Expressive One Word Picture Vocabulary Test-Fourth edition (EOWPVT) [48,49]. Standard scores from these assessments were used in analyses. 

Additionally, participants completed the Clinical Evaluation of Language Fundamentals-Fourth edition (CELF) [50]. From this measure, standard scores for core language, receptive language, and expressive language were utilized in analyses.

### 2.3. Procedures 

Participants traveled to Vanderbilt University Medical Center for each appointment; all measures were collected over the course of one to three visits. Parent surveys were provided to the parent or caregiver to complete during the first appointment. Language assessments were administered by either a certified speech-language pathologist or a student/clinical fellow under the supervision of a certified speech language pathologist. 

### 2.4. Analytic Plan

Prior to conducting analyses, missing data (ranging from 0–11% across variables derived for use in analyses) were imputed using the *missForest* package in R [51,52]. The first research question examined differences in patterns of sensory responsiveness between the autism and non-autism groups. Independent samples *t*-tests were performed on the aggregates of sensory responsiveness. 

The second research question examined correlations between aggregate scores for patterns of sensory responsiveness and the various indices of core and related features of autism. Bivariate correlational analyses, corrected for multiple comparisons using a Benjamini–Yekutieli false discovery rate correction, were utilized to evaluate the magnitude and direction of all associations of interest [53,54]. Steiger’s *z* tests were utilized to test the difference between bivariate correlations with dependent variables (i.e., indices of core and related autism features) in common using the *cocor* package in R [55,56]. Moderation analyses were conducted to test whether any associations of interest varied according to diagnostic group [57]. Cook’s D was utilized to monitor for undue influence throughout regression analyses; discrete data points with D values ≥ 1 were removed [58].

The third research question examined the proportion of variance in each core and related feature of autism jointly and uniquely attributable to each of the three patterns of sensory responsiveness. To answer this question, a series of multiple regression analyses were conducted. The aggregate variables for each pattern of sensory responsiveness were regressed onto each feature of autism in step-wise models utilizing systematically varied orders (see [59] for an example of this approach). The change in the proportion of variance accounted for in the final step of the model (Δ*R*^2^; e.g., the change in the variance accounted for when adding hyporesponsiveness into a model that already included sensory seeking and hyperresponsiveness as predictors) was considered the amount of unique variance attributable to each pattern of sensory responsiveness. To answer the fourth research question, each model containing two patterns of sensory responsiveness was then compared to the full model using the *anova* function in *R* to determine whether adding the third pattern of sensory responsiveness as a predictor resulted in a significant increase in model fit. 

## 3. Results

### 3.1. Group Differences

Results indicate that groups differed significantly in all three patterns of sensory responsiveness. On average, children with autism were rated as significantly higher in hyporesponsiveness (*t* = 13.12, *d* = 2.6, *p* < 0.001), hyperresponsiveness (*t* = 12.62, *d* = 2.5, *p* < 0.001), and sensory seeking (*t* = 7.77, *d* = 1.6, *p* < 0.001) compared to their peers without autism (see Figure 1). All effects were large in magnitude. These group differences are robust to controlling for nonverbal IQ (all *p* values for the effect of group remain <0.001).

### 3.2. Correlational Analyses

See Table 3 for a summary of findings for associations of interest for patterns of sensory responsiveness with core and related features of autism.

Hyporesponsiveness and hyperresponsiveness were not significantly associated with nonverbal IQ or age (*r* values = −0.10–−0.25). Increased sensory seeking, however, was negatively associated with age (*r* = −0.31, *p* = 0.009) and nonverbal IQ (*r* = −0.33, *p* = 0.004). Additionally, all three patterns of sensory responsiveness were strongly intercorrelated, *r* values = 0.57–0.79. 

All three patterns of sensory responsiveness were strongly correlated with the variables quantifying the core features of autism, the social communication and RRB subscales of the SRS and the core autism feature aggregate, with large magnitudes (*r* values = 0.67–0.78; see Figure 2A–C for an example). Steiger’s *z* tests indicated that the correlations between the social communication index of the SRS and the total autism feature aggregate with both hyperresponsiveness and hyporesponsiveness were stronger than those with sensory seeking (*z* values ≥ 3.07; *p* values ≤ 0.001). The remaining correlations for patterns of sensory responsiveness with core autism features were not significantly different.

In regards to adaptive behaviors, all three patterns of sensory responsiveness negatively correlated with the indices of the VABS with moderate to large magnitudes (*r* values = −0.44–−0.80), such that increased sensory behaviors were associated with decreased adaptive behaviors. The correlations for the VABS variables with hyporesponsiveness were consistently stronger than those with sensory seeking (*z* values ≥ 2.45; *p* values ≤ 0.014). The correlations for the socialization and daily living skills indices with hyperresponsiveness were also stronger than those with sensory seeking (*z* values = 2.94 and 2.57, respectively; *p* values = 0.003 and 0.010). The remaining correlations were not significantly different.

All of the indices from the BASC were moderately to strongly associated with hyperresponsiveness (*r* values = 0.45–0.81), such that increased hyperresponsiveness was associated with more parent report of internalizing behaviors and functional impairment; all of these correlations were stronger than those with sensory seeking (*r* values = 0.19–0.65; *z* values ≥ 2.69). Hyporesponsiveness was also moderately to strongly correlated with BASC scores (*r* values = 0.36–0.80); however, several of these correlations (i.e., those with anxiety and internalizing problem behaviors) were weaker than those with hyperresponsiveness (*z* values = 3.27 and 2.63, respectively; *p* values = 0.001 and 0.009).

In regards to language, all of the correlations with hyporesponsiveness (*r* values = −0.31–−0.41) and sensory seeking (*r* values = −0.32–−0.48) were moderate in magnitude, such that increased hyporesponsiveness and sensory seeking behavior was associated with decreased language (see Figure 2D,E for an example). The correlations between indices of language and hyperresponsiveness were small to moderate (*r* values = 0.27–0.37). None of the correlations between aggregate scores for patterns of sensory responsiveness and language significantly differed. None of the associations of interest in the study were moderated by group.

Given that groups differed on nonverbal IQ, post-hoc analyses were conducted in R to control for this factor [60]. Correcting for multiple comparisons, the majority of significant correlations between patterns of sensory responsiveness and the core and related features of autism were robust to controlling for nonverbal IQ (see Appendix A). When controlling for nonverbal IQ, however, selected relations between indices of sensory responsiveness and language were attenuated and no longer surpassed the threshold for statistical significance. Associations between patterns of sensory responsiveness and broad language ability as measured by the CELF, as well as the association between sensory seeking and receptive vocabulary as measured by the ROWPVT remained statistically significant when controlling for IQ (refer to Appendix A for further detail).

### 3.3. Unique Variance Explained by Patterns of Sensory Responsivness

See Table 4 for information regarding the amount of unique variance in each core and related feature of interest attributable to each pattern of sensory responsiveness and the total amount of variance accounted for by all three patterns in the final model.

In regards to the core features of autism, hyperresponsiveness and hyporesponsiveness both accounted for a significant amount of unique variance when added to the multiple regression model for all three dependent variables, while sensory seeking accounted for a significant amount of unique variance when added to the models for RRBs and core autism features. 

For indices derived from the VABS, hyporresponsiveness accounted for a significant amount of unique variance for all three domains, sensory seeking accounted for significant unique variance in communication, and hyperresponsiveness accounted for significant unique variance in daily living skills and socialization. Hyperresponsiveness was the only pattern of sensory responsiveness that significantly accounted for unique variance in the anxiety, depression, somatization, and internalizing problem behavior indices of the BASC. All three patterns of sensory responsiveness, however, significantly accounted for unique variance in the functional impairment probability index. 

Sensory seeking significantly accounted for unique variance in the expressive vocabulary (i.e., EOWPVT) standard scores and all three indices from the CELF. None of the other patterns of sensory responsiveness accounted for unique variance in language scores. 

## 4. Discussion

The present study sought to examine sensory responsiveness in older children and adolescents and its relation to the core and related features of autism. The older children and adolescents with autism in the present study exhibited significantly altered patterns of sensory responsiveness relative to a control group of peers without autism matched on chronological age and sex. Effect sizes for between-group differences across hyporesponsiveness, hyperresponsiveness, and sensory seeking were all large in magnitude. This finding extends the extant literature on sensory differences in individuals with autism by examining these patterns in a relatively understudied age-range using multiple previously developed and validated measures of sensory responsiveness with a large sample size. The robust differences between groups that we observed suggest that sensory differences observed early in life in individuals with autism do not resolve within this age range and are robust to controlling for nonverbal IQ. 

Our study is the first to comprehensively evaluate correlations between patterns of sensory responsiveness and the wide range of core and related symptoms of autism within a single sample. Even after correcting for multiple comparisons, nearly all of the correlations between patterns of sensory responsiveness and overall presence of core autistic features, adaptive behavior skills, internalizing problem behaviors, and language abilities were significant, with moderate to large effect sizes (the sole exception being the relation between somatization and sensory seeking, which was small and non-significant). The majority of these associations were also robust to controlling for nonverbal IQ (see Appendix A).

Sensory hyporesponsiveness and hyperresponsiveness were consistently and robustly correlated with the core features of autism, social communication differences in particular, and adaptive behavior. Of the three patterns of sensory responsiveness, hyporesponsiveness has been reported to be most prevalent in, and specific to, autism [4,24]. It has been proposed that hyporesponsiveness may be particularly disruptive to development, as children with high hyporesponsiveness often fail to orient towards, and may subsequently fail to engage with and learn from, sensory information in their environment [17,32]. Although hyperresponsiveness is also commonly observed in other neurodevelopmental conditions (e.g., attention deficit hyperactivity disorder [61,62]), hyperresponsiveness is also very frequently observed in autism [4] and has been proposed to be a causal factor in autism [63]. Notably, all three patterns of sensory responsiveness were strongly correlated with and accounted for unique variance in RRBs. Longitudinal work is much needed to determine whether alterations in patterns of sensory responsiveness precede broader differences in young children with or at increased likelihood for autism.

Interestingly, sensory hyperresponsiveness was most strongly associated with internalizing problem behaviors in this sample. This is, perhaps, not surprising, given that hyperresponsiveness has previously been linked with anxious behaviors [8,26]. It may be important for occupational therapists to intervene with children demonstrating hyperresponsiveness in order to reduce distress and curtail downstream consequences of internalizing problems (e.g., suicide, which may be increased in autism [64,65]) in this population [66]. 

The current study is the first to demonstrate robust links between language and sensory responsiveness in older children and adolescents with autism. All three patterns of sensory responsiveness displayed significant associations with our five variables indexing language ability. Of the three patterns of sensory responsiveness, sensory seeking tended to account for the greatest amount of unique variance in language-related constructs. This finding is consistent with the previous literature on young children and preschoolers with autism, which has consistently linked sensory seeking to language and communication abilities [16,17,20,21,25]. Future research should examine the emergence of sensory seeking behaviors as they relate to different aspects of language development (e.g., semantics, syntax) early in life. Additionally, given that language abilities are critical to literacy, e.g., [59], future research should examine the degree to which sensory responsiveness relates to reading proficiency in persons on the spectrum.

There are several strengths to the present study. First, we comprehensively examined between- group differences and associations of interest in a large sample of children with autism and a control group matched on several factors, including biological sex and chronological age. Diagnostic status of children within the autism group was confirmed via gold-standard measures, and all participants were well characterized. Multiple measures of sensory responsiveness were employed, increasing the stability, and thus the potential construct validity, of our indices of hyporesponsiveness, hyperresponsiveness, and sensory seeking [67]. Additionally, this study investigated correlations between sensory responsiveness and a broad range of core and related features of autism as evaluated via a very comprehensive battery of assessments. 

There are, however, limitations to this study. First, our sample was limited to older children and adolescents on the autism spectrum. Additional work is therefore necessary to ascertain the degree to which the present findings generalize to younger children and/or adults with autism. Additionally, we did not exclude participants from either group who had certain co-occurring conditions common to autism, such as depression, anxiety, and attention deficit hyperactivity disorder (ADHD). Given that other populations are known to exhibit these altered patterns of sensory responsiveness (e.g., individuals with ADHD [61,62], Down syndrome [68,69], specific language impairment [70]), additional work is warranted to determine whether the relations observed and reported here are specific to autism, or apply to neurodevelopmental conditions more broadly. Finally, we did not track the medications used by participants in the present study. Some medications that are prescribed to children with autism (e.g., psychotropic medications [71]) lead to altered brain function in this population e.g., [72]. Future studies should thus (a) track medication use and (b) evaluate whether there is an effect of psychotropic medication on sensory functioning in this population.

## 5. Conclusions

This study evaluated sensory responsiveness and its relation to clinical features in older children and adolescents with autism. We found that children with autism presented with significantly altered sensory responsiveness compared to matched peers without autism. Across both groups, all three patterns of sensory responsiveness were associated with core and related features of autism. Hyporesponsiveness and hyperresponsiveness were most robustly correlated with the core features of autism, in particular social communication differences, and adaptive behavior; all three patterns of sensory responsiveness were highly correlated with RRBs; hyperresponsiveness was most strongly associated with internalizing problem behaviors; and sensory seeking was most strongly associated with measures of language. Future research should investigate these relations longitudinally in young children with or at increased likelihood for autism and cross-sectionally in adults with autism and children with other neurodevelopmental conditions, such as Down syndrome and specific language impairment.

## Figures and Tables

**Figure 1 brainsci-10-00775-f001:**
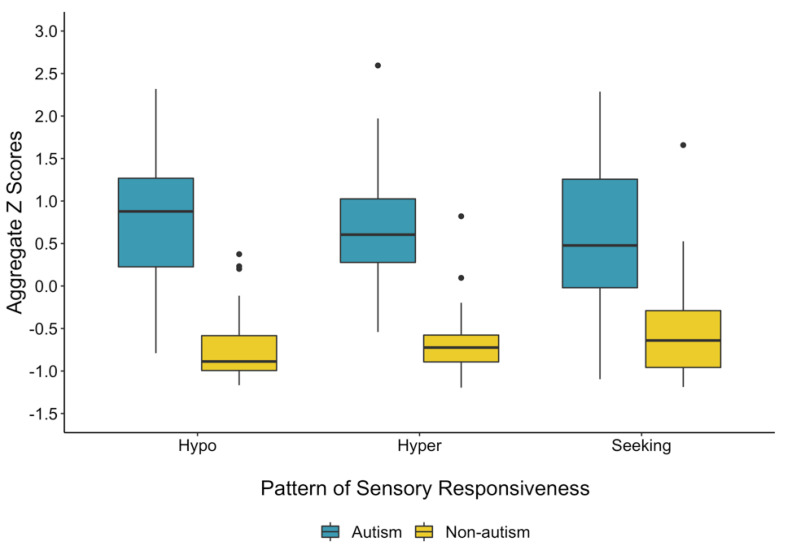
Group differences in sensory responsiveness. On average, children with autism (blue) were rated as significantly higher than the children without autism (yellow) in all three patterns of sensory responsiveness. Hypo = hyporesponsiveness; hyper = hyperresponsiveness; seeking = sensory seeking.

**Figure 2 brainsci-10-00775-f002:**
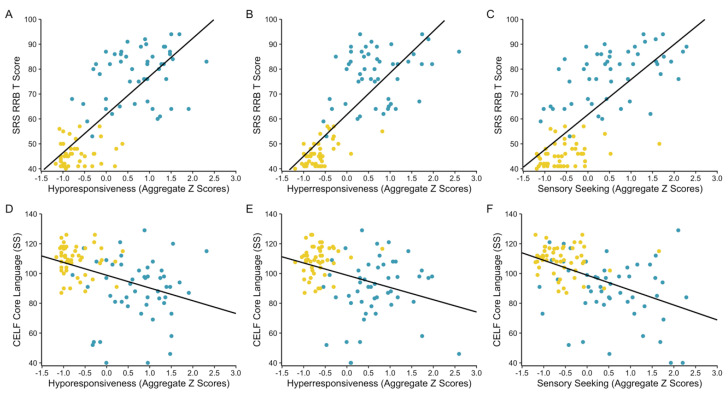
Scatterplots depicting selected associations between patterns of sensory responsiveness and features of autism. All three patterns of sensory responsiveness were positively correlated with the restricted, repetitive behaviors (RRB) index T scores from the Social Responsiveness Scale (SRS; (**A**–**C**)) [45] and negatively correlated with core language standard scores (SS) from the Clinical Evaluation of Language Fundamentals-Fourth edition (CELF; (**D**–**F**)) [50]. All depicted associations were statistically significant; effects of interest were observed across, and did not significantly vary according to, diagnostic group (autism group plotted in blue; non-autism group plotted in yellow).

**Table 1 brainsci-10-00775-t001:** Means and standard deviations of selected variables by group

	Autism (*n* = 50) *M* (*SD*) or *n* (%)	Non-Autism (*n* = 50) *M* (*SD*) or *n* (%)
Chronological Age (Years)	13.1 (3.1)	13.0 (2.7)
Nonverbal IQ *	109.0 (16.2)	118.0 (13.0)
Biological Sex		
Male	37 (74%)	37 (74%)
Female	13 (26%)	13 (26%)
Race		
Asian	3 (6%)	1 (2%)
Black or African American	2 (4%)	4 (8%)
White	35 (70%)	40 (80%)
Multiple Races	7 (14%)	5 (10%)
Not Reported	3 (6%)	0 (0%)
Ethnicity		
Hispanic or Latino	5 (10%)	5 (10%)
Not Hispanic or Latino	43 (86%)	44 (88%)
Not Reported	2 (4%)	1 (2%)

Note: Nonverbal IQ = Nonverbal intelligence as measured by the Leiter International Performance Scale–Third edition or the Test of Nonverbal Intelligence–Fourth edition [40,41]. * Denotes groups significantly differed, *p* < 0.01.

**Table 2 brainsci-10-00775-t002:** Summary of measures and variables derived for use in analyses

Measure	Type	Variables Used in Analyses
Measures of Nonverbal Cognitive Ability		
Leiter International Performance Scale–Third edition (Leiter–3) [41] or Test of Nonverbal Intelligence–Fourth edition (TONI–4) [40]	Standardized Assessment	Nonverbal IQ
Measures of Sensory Responsiveness		
Sensory Experiences Questionnaire-Version 2.1 [5]	Parent Report	Sensory Seeking ^a^, Hyperresponsiveness ^b^, and Hyporesponsiveness ^c^ mean scores
Sensory Profile [44]	Parent Report	Low Registration (Hypo) ^c^, Sensation Seeking (Seeking) ^a^, Sensory Sensitivity (Hyper) ^b^, and Sensation Avoiding (Hyper) ^b^ scores
Measures of Core Features of Autism		
Social Communication Questionnaire [43]	Parent Report	Total score^d^
Social Responsiveness Scale [45]	Parent Report	Restricted Interests and Repetitive Behavior, Social Communication Impairment, and Overall Autistic Features ^d^ T-scores
Measures of Related Features of Autism		
Vineland Adaptive Behavior Scales–Second edition [46]	Parent Report	Communication, Daily Living Skills, and Socialization Domain standard scores
Behavior Assessment System for Children–Third edition [47]	Parent Report	Anxiety, Depression, and Somatization scale T-scores; Internalizing Problem Behaviors and Functional Impairment Probability Index T-scores
Measures of Language Ability		
Receptive One Word Picture Vocabulary Test–Fourth edition [48]	Standardized Assessment	Receptive Vocabulary standard score
Expressive One Word Picture Vocabulary Test–Fourth edition [49]	Standardized Assessment	Expressive Vocabulary standard score
Clinical Evaluation of Language Fundamentals–Fourth edition [50]	Standardized Assessment	Core Language, Receptive Language and Expressive Language Index standard scores

Note: Seeking = Sensory Seeking, Hyper = Hyperresponsiveness, Hypo = Hyporesponsiveness. Variables were aggregated to measure: ^a^ sensory seeking, ^b^ hyperresponsiveness, ^c^ hyporesponsiveness, ^d^ core autism features.

**Table 3 brainsci-10-00775-t003:** Zero-order correlations between patterns of sensory responsiveness and core and related features of autism

	Hypo	Hyper	Seeking
Sample Characteristics			
Age	−0.10	−0.12	−0.31 **
Nonverbal IQ	−0.23	−0.25	−0.33 **
Sensory Responsiveness			
Hypo	1	0.79 ***	0.60 ***
Hyper	0.79 ***	1	0.57 ***
Seeking	0.60 ***	0.57 ***	1
Core Features of Autism			
SRS Social Communication T score	0.85 ***	0.83 ***	0.62 ***
SRS RRB T-score	0.78 ***	0.80 ***	0.72 ***
Core Autism Feature Aggregate	0.87 ***	0.83 ***	0.67 ***
Related Features of Autism			
VABS Communication SS	−0.72 ***	−0.65 ***	−0.57 ***
VABS Daily Living Skills SS	−0.66 ***	−0.63 ***	−0.44 ***
VABS Socialization SS	−0.80 ***	−0.75 ***	−0.57 ***
BASC Anxiety T score	0.44 ***	0.61 ***	0.29 *
BASC Depression T score	0.63 ***	0.70 ***	0.52 ***
BASC Somatization T score	0.36 **	0.45 ***	0.19
BASC Internalizing Problem Behaviors T score	0.57 ***	0.70 ***	0.40 ***
BASC Functional Impairment Probability T score	0.80 ***	0.81 ***	0.65 ***
Language			
ROWPVT SS	−0.32 **	−0.28 *	−0.40 ***
EOWPVT SS	−0.31 **	−0.27 *	−0.32 **
CELF Receptive Language SS	−0.38 **	−0.35 **	−0.44 ***
CELF Expressive Language SS	−0.39 ***	−0.35 **	−0.45 ***
CELF Core Language SS	−0.41 ***	−0.37 **	−0.48 ***

Note: Hypo = hyporesponsiveness aggregate score, Hyper = hyperresponsiveness aggregate score, Seeking = sensory seeking aggregate score, Nonverbal IQ = Nonverbal intelligence as measured by the Leiter International Performance Scale-Third edition [41] or the Test of Nonverbal Intelligence-Fourth edition [40], SRS = Social Responsiveness Scale [45], Core Autism Feature Aggregate = Aggregate of SRS total T score and Social Communication Questionnaire total score [43], SS = Standard score, VABS = Vineland Adaptive Behavior Scales-Second edition [46], BASC = Behavior Assessment System for Children-Third edition [47], ROWPVT = Receptive One Word Picture Vocabulary Test-Fourth edition [48], EOWPVT = Expressive One Word Picture Vocabulary Test-Fourth edition [49], CELF = Clinical Evaluation of Language Fundamentals-Fourth edition [50]. * *p* < 0.05, ** *p* < 0.01, *** *p* < 0.001 after applying a Benjamini–Yekutieli correction [53].

**Table 4 brainsci-10-00775-t004:** Variance in each measure of core and related features of autism explained jointly and uniquely by each pattern of sensory responsiveness

	Unique Variance			Total Variance Explained
	Hypo	Hyper	Seeking	
Core Features of Autism				
SRS Social Communication T score	8.57% ***	4.82% ***	0.64%	79.91%
SRS RRB T score	2.53% **	5.28% ***	7.06% ***	76.65%
Core Autism Feature Aggregate	8.60% ***	3.93% ***	1.86% **	83.47%
Related Features of Autism				
VABS Communication SS	7.85% ***	0.93%	2.07% *	55.73%
VABS Daily Living Skills SS	5.99% **	2.90% *	0.04%	47.16%
VABS Socialization SS	8.88% ***	3.07% **	0.48%	68.11%
BASC Anxiety T score	0.41%	19.23% ***	0.20%	38.52%
BASC Depression T score	0.67%	9.25% ***	1.26%	52.12%
BASC Somatization T score	0.08%	8.03% **	0.80%	20.88%
BASC Internalizing Problem Behaviors T score	0.06%	15.52% ***	0.00%	48.40%
BASC Functional Impairment Probability T score	3.74% ***	6.57% ***	2.68% **	75.25%
Language				
ROWPVT SS	1.01%	0.01%	2.69%	12.58%
EOWPVT SS	0.76%	0.02%	6.22% ***	16.85%
CELF Receptive Language SS	0.81%	0.05%	6.42% **	21.43%
CELF Expressive Language SS	1.21%	0.00%	6.50% **	22.28%
CELF Core Language SS	1.04%	0.01%	8.22% **	25.69%

Note: Hypo = hyporesponsiveness aggregate score, Hyper = hyperresponsiveness aggregate score, Seeking = sensory seeking aggregate score, Nonverbal IQ = Nonverbal intelligence as measured by the Leiter International Performance Scale-Third edition [41] or the Test of Nonverbal Intelligence-Fourth edition [40], SRS = Social Responsiveness Scale [45], Core Autism Feature Aggregate = Aggregate of SRS total T score and Social Communication Scale total score [43], SS = Standard score, VABS = Vineland Adaptive Behavior Scales-Second edition [46], BASC = Behavior Assessment System for Children-Third edition [47], ROWPVT = Receptive One Word Picture Vocabulary Test-Fourth edition [48], EOWPVT = Expressive One Word Picture Vocabulary Test-Fourth edition [49], CELF = Clinical Evaluation of Language Fundamentals-Fourth edition [50]. * *p* < 0.05, ** *p* < 0.01, *** *p* < 0.001.

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
