# Peer review of "Relations between Sensory Responsiveness and Features of Autism in Children"

_brainsci, 2020, doi:10.3390/brainsci10110775_

Round 1
Reviewer 1 Report
Feldman et al studied relation between sensory responsiveness (hyporesponsiveness, hyperresponsiveness and sensory seeking) and features (core and related) of autism in older children and adolescents, and found strong association between hyperresponsiveness and internalizing problem behaviors, and between sensory seeking and language abilities.
In general, this study is well designed and conducted. The results are nicely and succinctly presented, and some of them are important.
In the Materials and Methods section, the authors should describe demographic data for the non-autism group in more detail: e.g. Were children with ADHD included? If yes, hoy many?
Author Response
Reviewer 1: In the Materials and Methods section, the authors should describe demographic data for the non-autism group in more detail: e.g. Were children with ADHD included? If yes, [how] many?
OUR RESPONSE: We have added additional demographics data (i.e., race, ethnicity) to our description of our participants in Table 1. We do not have detailed information on ADHD status in either group; we now acknowledge this as a limitation of the study.
Reviewer 2 Report
This is a well-written paper dealing with the ever-current topic of sensory abnormalities in individuals with autism spectrum disorder. Some comments follow.
- The authors mentioned among the inclusion criteria “no diagnosed genetic disorders, such as Fragile X or tuberous sclerosis” (line 105-106): which medical work-up was performed in order to exclude a genetic disorder?
- Figure 3 (line 234) is Figure 2.
- Why didn't the authors use the calibrated severity score of ADOS-2 in order to look for correlations between autism severity and sensory responsiveness?
- Were the participants, and particularly those with autism, receiving psychotropic drugs that can modify the symptoms of an altered sensoriality?
- I suggest to avoid expressions like “autistic individuals”: “individuals with autism” is more correct.
- It’s useful verify the references: American Psychiatric (not Psychological) Association for reference 1 and 2.
Author Response
The authors mentioned among the inclusion criteria “no diagnosed genetic disorders, such as Fragile X or tuberous sclerosis” (line 105-106): which medical work-up was performed in order to exclude a genetic disorder?
OUR RESPONSE: We changed this inclusion criterion to state "no parent report of diagnosed genetic disorders," as we did not obtain medical work-up related to genetic disorders.
Figure 3 (line 234) is Figure 2.
OUR RESPONSE: Thank you for bringing this to our attention; we corrected this error.
Why didn't the authors use the calibrated severity score of ADOS-2 in order to look for correlations between autism severity and sensory responsiveness?
OUR RESPONSE: We did not utilize ADOS calibrated severity scores for several reasons. First, we were interested in the correlations across groups, and only the participants in our autism group received the ADOS. Further, the ADOS provides a truncated range of scores, which impacts our ability to detect effects of interest (e.g., Huck, 1992). Finally, the SRS-2 is a reliable alternative that has high convergent validity and can be used with individuals with and without autism (e.g., Armstrong & Iarocci, 2013; Chan et al., 2017; Constantino et al., 2003; Wigham et al., 2012).
Given that Reviewer 2 is curious, we did analyze correlations between the ADOS-2 calibrated severity scores and the aggregates of sensory responsiveness.The correlation between ADOS-2 calibrated severity scores was significantly correlated with sensory seeking in the autism group, r(50) = .32, p = .023, but not with hyporesponsiveness, r(50) = .10, p = .50, or hyperresponsiveness, r(50) = -.01, p = .97.
Were the participants, and particularly those with autism, receiving psychotropic drugs that can modify the symptoms of an altered sensoriality?
OUR RESPONSE: That is an excellent point - we did not track the medication use of participants in our study. Therefore, we have added the following limitation and future direction: "We did not track the medications used by participants in the present study. Some medications that are prescribed to children with autism (e.g., psychotropic medications [71]) lead to altered brain function in this population [e.g., 72]. Future studies should (a) track medication use and (b) evaluate whether there is an effect of psychotropic medication on sensory functioning in this population."
I suggest to avoid expressions like “autistic individuals”: “individuals with autism” is more correct.
OUR RESPONSE: We thank Reviewer 2 for providing this feedback. As we state in the endnote of our paper, "there is some debate in the literature regarding whether researchers should utilize person-first language (e.g., individuals with autism, individuals with ASD) versus identity-first language (e.g., autistic individuals). Clinicians and researchers tend to prefer person-first language, while many autistic individuals and their allies prefer and advocate for identity-first language [73,74]. In this manuscript, we refer to children with autism using person-first language while otherwise avoiding ableist language [75], and acknowledge that this approach may not be received equally by all of the stakeholders of our research." We have amended our only usage of identity-first language (line 380) to now state "adults with autism."
It’s useful verify the references: American Psychiatric (not Psychological) Association for reference 1 and 2.
OUR RESPONSE: Thank you for bringing this to our attention; we corrected this error.